# Non-Viable *Lactobacillus johnsonii* JNU3402 Protects against Diet-Induced Obesity

**DOI:** 10.3390/foods9101494

**Published:** 2020-10-19

**Authors:** Garam Yang, Eunjeong Hong, Sejong Oh, Eungseok Kim

**Affiliations:** 1Department of Biological Sciences, College of Natural Sciences, Chonnam National University, Gwangju 61186, Korea; yanggaram@naver.com (G.Y.); okok44554@naver.com (E.H.); 2Division of Animal Science, College of Agriculture & Life Sciences, Chonnam National University, Gwangju 61186, Korea

**Keywords:** non-viable *Lactobacillus johnsonii* JNU3402, diet-induced obesity, peroxisome proliferator-associated receptor-γ, body temperature

## Abstract

In this study, the role of non-viable *Lactobacillus johnsonii* JNU3402 (NV-LJ3402) in diet-induced obesity was investigated in mice fed a high-fat diet (HFD). To determine whether NV-LJ3402 exhibits a protective effect against diet-induced obesity, 7-week-old male C57BL/6J mice were fed a normal diet, an HFD, or an HFD with NV-LJ3402 for 14 weeks. NV-LJ3402 administration was associated with a significant reduction in body weight gain and in liver, epididymal, and inguinal white adipose tissue (WAT) and brown adipose tissue weight in HFD-fed mice. Concomitantly, NV-LJ3402 administration to HFD-fed mice also decreased the triglyceride levels in the plasma and metabolic tissues and slightly improved insulin resistance. Furthermore, NV-LJ3402 enhanced gene programming for energy dissipation in the WATs of HFD-fed mice as well as in 3T3-L1 adipocytes with increased peroxisome proliferator-activated receptor-γ (PPARγ) transcriptional activity, suggesting that the PPARγ pathway plays a key role in mediating the anti-obesity effect of NV-LJ3402 in HFD-fed mice. Furthermore, NV-LJ3402 administration in HFD-fed mice enhanced mitochondrial levels and function in WATs and also increased the body temperature upon cold exposure. Together, these results suggest that NV-LJ3402 could be safely used to develop dairy products that ameliorate diet-induced obesity and hyperlipidemia.

## 1. Introduction

High-caloric intake and low physical activity promote excessive fat accumulation in metabolic tissues including adipose tissues, which is the main cause of obesity and associated metabolic disorders, such as insulin resistance, diabetes, nonalcoholic hepatic steatosis, hyperlipidemia, and cardiovascular disease [1].

Probiotics are living microorganisms that confer a health benefit to the host. Accumulating evidence shows that probiotics remodel the gut-microbiome and signaling pathways whose functioning is altered in response to metabolic imbalances, thereby helping to recover metabolic and immune functions [2]. Recent studies have shown that some *Lactobacillus* strains protect against diet-induced obesity in mice. *Lactobacillus acidophilus* NS1 inhibits obesity in high-fat diet (HFD)-fed mice by increasing hepatic fatty acid oxidation with decreased lipogenesis, improving HFD-induced insulin resistance, nonalcoholic fatty liver disease, and hyperlipidemia [3]. In addition, another *Lactobacillus* strain, *Lactobacillus amylovorus* KU4, promotes the browning of WAT in HFD-fed mice, leading to the suppression of HFD-induced adiposity with reduced lipid deposition in metabolic tissues, such as liver and WAT as well as improved insulin sensitivity [4]. The World Health Organization and the Food and Agriculture Organization of the United Nations define probiotics as living beneficial microorganisms [5]. However, several studies have recently reported that nonliving microorganisms may also have beneficial health effects, such as enhanced immune function and reduced diarrhea [6]. Therefore, non-viable (NV) probiotic bacteria have been garnering increasing interest by virtue of their health benefits, which are known to be similar to those of regular probiotics [7]. Furthermore, NV probiotics are not encumbered by risks associated with live microorganisms and are easier to handle than regular probiotics [7]. Contrary to popular belief, the dairy industry has leveraged the concept of NV probiotics for centuries. The availability of dairy products and other foods increases upon using NV probiotics. Although inactivated starter organisms are used in yogurt due to their ability to increase the shelf life [8], they are empirically understood to have unknown health effects.

*Lactobacillus johnsonii* was first classified with *L. acidophilus* and is often isolated from the intestines of humans and animals [9]. Probiotic characteristics are presented by various *L. johnsonii* strains, including the inhibition of pathogens in the gut, alleviation of diabetes symptoms, reduction of serum cholesterol levels, immune-stimulation, and adhesion to intestinal epithelial cells [10,11,12]. *L. johnsonii* JNU3402 (LJ3402) isolated from Korean infant feces exhibit bile and acid resistance. The protective effects of NV *Lactobacillus* strains against diet-induced obesity have largely remained unexplored. Therefore, the effect of NV-LJ3402 on diet-induced obesity was determined using HFD-fed mice. This study demonstrated that NV-LJ3402 enhanced the expression of the metabolic genes involved in energy expenditure, partly by stimulating the proliferator-associated receptor-γ (PPARγ) activity and mitochondrial levels in WAT, increasing the body temperature and resulting in protection from diet-induced obesity.

## 2. Materials and Methods

### 2.1. Preparation of NV-LJ3402

NV-LJ3402 was prepared using the following method. LJ3402 cells were cultured in MRS broth (BD, Difco Laboratories, Detroit, MI, USA) at 37 °C for 24 h and then the cells were centrifuged to 4000× *g* for 15 min at 4 °C. After that, the pellets were then washed 3 times with sterile Phosphate-buffered saline (PBS, 0.01 M, pH 7.2) and resuspended to reach a density of ca. 1 × 10^8^ or 1 × 10^9^ cfu/mL. Normalized cells were killed at 80 °C for 15 min in a water bath and the lack of bacterial colonies were confirmed using MRS agar (BD, Difco Laboratories, Detroit, MI, USA) plates.

### 2.2. Animals

Six-week-old C57BL/6J male mice (weight, 19–20 g, Central Animal Laboratory, Daejeon, Korea) were acclimated for 1 week and then fed either a normal diet (16% of total calories from fat, LabDiet, St. Louis, MO, USA) or HFD (45% of total calories from fat, Research Diets Inc., New Brunswick, NJ, USA). Two-hundred microliters of NV-LJ3402 resuspension or PBS was orally administered daily to the mice for 14 weeks. All animal procedures were approved by the Institutional Animal Care and Use Committees at Chonnam National University (CNU IACUC-YB-2019-06, Approved date: 19 February 2019).

### 2.3. Cell Culture and Transfection

HEK293T and 3T3-L1 cells were cultured in Dulbecco’s Modified Eagle Medium containing 5% fetal bovine serum or 10% newborn calf serum and antibiotics. Plasmids, pGL3-UCP1 promoter (-2620 to +68 bp), pGL3-ACOX-PPRE-Luc, pGL3-CETP-LXRE-Luc, pGL3-TK-IR-1-Luc, pCDNA3-PPARγ, pCMX-RXRα, pCMX-LXR, and pSG5-FXR have been described previously [3,13,14,15]. MRS broth was heated at 80 °C for 15 min and used as a control bacterial culture medium (con). NV-LJ3402-CM or con was added to 3T3-L1 adipocytes on day 6 for 48 h at 1/100 volume of the medium to test the effect of NV-LJ3402-CM on gene expression, mitochondrial levels, and lipid accumulation. In addition, NV-LJ3402-CM was added to HEK293T cells for 24 h to test its effect on the activities of transcription factors. Adipocyte differentiation of 3T3-L1 cells and transfections were performed as previously described [4].

### 2.4. Plasma and Tissue Analyses

The plasma and tissue levels of triglyceride (TG) and insulin were measured using a TG quantification kit (SCG Biomax, Seoul, Korea) and insulin ELISA (ALPCO, Salem, NH, USA), respectively, according to the manufacturer’s protocol. After 8 h of fasting, mice were injected intraperitoneally with glucose (1.5 g/kg BW) for the glucose tolerance test. Blood glucose levels were measured by tail bleeding.

### 2.5. Analyses of Mitochondrial DNA and Citrate Synthase Activity

DNA was isolated from WATs and 3T3-L1 adipocytes using a genomic DNA isolation kit (Qiagen, Valencia, CA, USA or GeneAll biotechnology, Seoul, Korea) and quantitative PCR was performed to determine mitochondrial DNA (mtDNA) copy number using mtDNA primers and nuclear DNA primers. The fold ratio of mitochondrial DNA levels relative to nuclear DNA was calculated. Citrate synthase activity was measured in WATs and 3T3-L1 adipocytes using a citrate synthase activity assay kit (BioVision, Milpitas, CA, USA).

### 2.6. RNA Isolation, Reverse Transcription, and RT-qPCR

Total RNA was isolated from 3T3-L1 cells and WATs using Trizol reagent (Invitrogen, Waltham, MA, USA), and cDNA was synthesized from 1 μg total RNA using Moloney Murine Leukemia Virus (M-MLV)Reverse Transcriptase (Promega, Madison, WIS, USA). Real time-quantitative PCR (RT-qPCR) was performed as previously described [4], and the results were normalized to 36B4 mRNA expression. Relative quantification of PCR products was calculated by the difference in Ct values between the target and 36B4 genes using the 2^−ΔΔCt^ method. Primer sequences for PCR are listed in Table 1.

### 2.7. Yogurt Fermentation

Homogenized whole milk (3.4% fat, 8.5% milk solid-non-fat, SNF) and skim milk powder (0.1% fat, 95% SNF) were obtained from a local dairy plant (Seoul Dairies, Seoul, Korea). Skim milk powder was added at a level of 2.5% to increase milk solids (11% SNF). One liter of milk base was heated at 95 °C in a glass bottle (Schott Duran, Germany) for 10 min and then cooled to 42 °C. Yogurt starters were inoculated with *Streptococcus thermophilus* (ca. 5 × 10^6^ cfu/mL, Chr. Hansen Holdings A/S, Hoersholm, Denmark) and *L. delbruekii* subsp. *bulgaricus* (ca. 5 × 10^6^ cfu/mL, Chr. Hansen Holdings A/S). After that, NV-LJ3402 (10^8^ and 10^9^) was added to the experimental group and then fermented at 42 °C until the pH reached 4.5 [16].

### 2.8. Viable Cells and pH Measurements

The pH values were monitored using a Multi-Channel pH/Ion meter with a temperature probe (PhysioLab, Pusan, Korea) and recorded every 60 s during the fermentation process. Viable cells of *L. delbrueckii* subsp. *bulgaricus* were enumerated using MRS agar (BD, Difco Laboratories, Detroit, MI, USA) adjusted to pH 5.4 and incubated anaerobically at 37 °C for 48 h. For enumeration of *S. thermophilus*, diluted samples were incubated at 43 °C for 24 h using M17 agar (BD).

### 2.9. Statistical Analysis

All values are expressed as mean ± S.E.M. Statistical significance of data was determined by Student’s *t*-test for comparisons between two groups, namely, ND vs. HFD groups, HFD vs. HFD-NV-LJ3402 groups in mice, con vs. NV-LJ3402-CM groups, and PPARγ vs. PPARγ+NV-LJ3402-CM groups in 3T3-L1 adipocytes. Statistical differences in gene expression and lipid accumulation among different treatment groups in 3T3-L1 adipocytes were analyzed by Tukey’s multiple comparison test using SAS software (Version 9.4, SAS Institute, Cary, NC, USA). *p* less than 0.05 were considered statistically significant.

## 3. Results

### 3.1. Effect of NV-LJ3402 on HFD-Induced Body Weight Gain and Adiposity in Mice

Recent studies in mice and human volunteers showed that several strains of probiotic bacteria, such as *Lactobacillus acidophilus* NS1, *Lactobacillus amylovorus* KU4, and *Akkermansia muciniphila*, ameliorate obesity and associated metabolic disorders [3,4,17]. To determine whether NV probiotic bacteria have a similar effect on the reduction of diet-induced obesity, 7-week-old male mice were fed HFD with or without oral administration of NV-LJ3402 for 14 weeks. Administration of NV-LJ3402 to HFD-fed mice (NV-LJ3402 mice) reduced HFD-induced body weight gain by 10% (HFD, 39.79 ± 1.04 g vs. NV-LJ3402, 35.95 ± 1.01 g, *p* < 0.05) in parallel with a reduction in liver weight (14%, HFD, 1.63 ± 0.07 g vs. NV-LJ3402, 1.37 g ± 0.04 g, *p* < 0.05), epididymal WAT (eWAT, 26%, HFD, 1.24 ± 0.05 g vs. NV-LJ3402, 0.92 g ± 0.05 g, *p* < 0.01), inguinal WAT (iWAT, 30%, HFD, 1.27 ± 0.09 g vs. NV-LJ3402, 0.90 g ± 0.06 g, *p* < 0.01), and brown adipose tissue weight (30%, HFD, 0.21 ± 0.01 g vs. NV-LJ3402, 0.15 g ± 0.01 g; *p* < 0.01), compared to those in HFD-fed mice (HFD mice) (Figure 1a,c). However, there was no significant difference in food intake between these two groups (Figure 1b). Furthermore, TG levels in eWAT and liver in NV-LJ3402 mice were reduced by 36% (34.87 ± 0.74 mg/g eWAT, *p* < 0.01) and 70% (8.84 ± 1.60 mg/g liver, *p* < 0.01), respectively, as compared with those (55.43 ± 3.01 mg/g eWAT and 17.09 ± 1.27 mg/g liver) in HFD-fed mice (HFD mice) (Figure 1d). As expected, plasma levels of TG were distinctly increased in response to HFD. However, when NV-LJ3402 was administered during HFD feeding, plasma TG levels in HFD mice were reduced by 20% (HFD, 1.71 ± 0.05 mmol/L vs. NV-LJ3402, 1.33 ± 0.07 mmol/L, *p* < 0.01), indicating that the NV-LJ3402 administration confers resistance to HFD-induced obesity and lipid accumulation in the metabolic tissues. Since diet-induced obesity is closely associated with insulin resistance, the possible effect of NV-LJ3402 on insulin resistance was determined. As expected, HFD increased the postprandial and fasting plasma glucose levels by 1.5-fold and 2-fold, respectively, compared to those in ND-fed mice *(p* < 0.01, Figure 1e). However, when NV-LJ3402 was administered to HFD mice, fed and fasting plasma glucose levels and plasma insulin levels were reduced to 26% (*p* < 0.05), 40% (*p* < 0.05), and 51% (*p* < 0.05), respectively, when compared to those in HFD mice. Next, glucose tolerance was compared between HFD and NV-LJ3402 mice. As expected, intraperitoneal glucose injections increased plasma glucose levels in both HFD and NV-LJ3402 mice to a similar level (Figure 1e). The area under the curve (AUC) analysis for the glucose tolerance test showed that NV-LJ3402 mice exhibited slightly lower glucose levels than HFD mice. However, the glucose-lowering effect of NV-LJ3402 was not statistically significant (*p* = 0.0934 vs. HFD). NV-LJ3402 administration reduced plasma glucose levels after 45 min after glucose injection. This NV-LJ3402-mediated reduction in plasma glucose levels became significant 90 min after glucose injection, while the plasma glucose levels in HFD-fed mice did not significantly change until this time.

### 3.2. Effect of NV-LJ3402 on the Expression of Metabolic Genes in the WAT of HFD Mice and 3T3-L1 Adipocytes

HFD leads to the dysregulation of genes involved in lipid metabolism in metabolic tissues such as WAT, and these alterations in the metabolic gene profile are closely associated with obesity and related metabolic diseases [18]. Therefore, to identify the metabolic pathway underlying the protective effect of NV-LJ3402 against HFD-induced obesity, RT-qPCR was performed to determine the expression profile of genes coding for proteins involved in lipid metabolism in the WAT. NV-LJ3402 reduced the mRNA levels of lipogenic genes, such as fatty acid synthase (FAS), acetyl-CoA carboxylase (ACC), and sterol regulatory element-binding protein-1c (SREBP1c), compared with those in eWAT of HFD mice (*p* < 0.05). Conversely, NV-LJ3402 increased the mRNA levels of genes coding for proteins involved in beta-oxidation, such as acetyl-CoA oxidase (ACOX), carnitine palmitoyltransferase 1 (CPT1), and peroxisome proliferator-activated receptor gamma coactivator 1-α (PGC1α) in eWAT of HFD mice (*p* < 0.05 vs. HFD, Figure 2a). Furthermore, NV-LJ3402 mice also exhibited increased mRNA levels of browning genes in iWAT (*p* < 0.05), such as uncoupling protein 1 (UCP1), PPARγ, and Cidea when compared with those in eWAT of HFD mice. Additionally, when day 6 3T3-L1 adipocytes (day 6 after differentiation) were treated with NV-LJ3402-CM for 48 h, the mRNA levels of genes coding for proteins regulating energy dissipation (UCP1, PPARγ, Cidea, ACOX, CPT1, and PGC1α) were significantly increased (*p* < 0.05 or *p* < 0.01 vs. con) in parallel with the reduction in mRNA levels of lipogenic genes (FAS, ACC, and SREBP1c, *p* < 0.05 vs. con, Figure 2b). Since NV-LJ3402 showed a regulatory effect with respect to the expression of genes involved in lipid metabolism, we next analyzed whether NV-LJ3402 could inhibit lipid accumulation in 3T3-L1 adipocytes by Oil Red O staining. When 3T3-L1 adipocytes were treated with NV-LJ3402-CM, lipid accumulation on day 8 progressively reduced depending on the treatment duration (22%, 46%, and 61% decrease after 24, 36, and 48 h treatment, respectively), compared to that in the control adipocytes (*p* < 0.01 or *p* < 0.001, Figure 2c). Together, these results suggest that NV-LJ3402 might reduce lipid accumulation by regulating genes coding for proteins involved in lipid metabolism and browning, leading to reduced adiposity in HFD mice.

### 3.3. NV-LJ3402 Regulates the Expression of Metabolic Genes by Promoting PPARγ Transcriptional Activity

NV-LJ3402 has been shown to alter the expression of genes coding for proteins involved in lipid metabolism in the WAT of HFD mice and 3T3-L1 adipocytes. To determine whether NV-LJ3402 regulates the expression of these metabolic genes by controlling the activity of transcription factors that play a key role in adipocyte biology, reporter gene assays were performed in HEK293T cells using reporter genes harboring binding sites of these transcription factors. As expected, all transcription factors tested were activated by their cognate ligands (Figure 3a). However, while NV-LJ3402-CM increased the PPARγ transcriptional activity, NV-LJ3402-CM did not significantly affect the transcriptional activities of liver X receptor and farnesoid X receptor, suggesting that NV-LJ3402-CM specifically enhances PPARγ activity to mediate its effect on the expression of genes involved in lipid metabolism. Next, to further confirm the induction of PPARγ transcriptional activity by NV-LJ3402, two other reporter genes harboring the promoter element of either PPARγ or UCP1 were used. As shown in Figure 3b, NV-LJ3402-CM increased PPARγ activity by approximately 2.0‒2.3-fold, depending on the reporter genes, relative to PPARγ alone (*p* < 0.05). In addition, similar to the effect of NV-LJ3402-CM, CM from live LJ3402 also enhanced PPARγ activity. Furthermore, RT-qPCR showed that the mRNA levels of PPARγ and PPARγ target genes (ACOX and UCP1) were increased in day 8 3T3-L1 adipocytes treated with NV or live LJ3402-CM for 48 h (*p* < 0.05 vs. con, Figure 3c). However, on day 6, 3T3-L1 adipocytes were co-treated with LJ3402-CM and 10 μM GW9662, a PPARγ-specific antagonist, for 48 h, the enhancing effect of LJ3402-CM on the expression of these genes was not observed, suggesting that LJ3402-CM may increase the expression of genes coding for proteins involved in energy dissipation by enhancing the PPARγ activity. Recent studies have shown that various thermogenic stimuli such as β-adrenergic stimulation, cold exposure, and PPARγ agonist treatment could trigger brown-like adipocytes in subcutaneous iWAT, leading to the suppression of fat accumulation in metabolic tissues by increasing the dissipation of energy as heat. PPARγ plays a key role in this process by upregulating the expression of UCP1, which is a key brown adipose tissue marker. As the NV-LJ3402-induced an increase in UCP1 and Cidea expression by enhancing PPARγ activity was already observed in the subcutaneous iWAT of HFD mice and 3T3-L1 adipocytes, the potential inhibitory effect of GW9662 on the NV-LJ3402-induced reduction in lipid accumulation in 3T3-L1 adipocytes was determined using Oil Red O staining. As expected, 48-h treatment of day 6 3T3-L1 adipocytes with a browning stimulus (100 μM isoproterenol), live LJ3402-CM, and NV-LJ3402-CM reduced lipid accumulation by 70%, 67%, and 68%, respectively, compared with that in the control adipocytes (*p* < 0.05, Figure 3d). However, when cells were treated with GW9662 in the background of isoproterenol or LJ3402-CM treatment, the suppressive effect of isoproterenol and NV-LJ3402-CM on lipid accumulation was partially attenuated.

### 3.4. NV-LJ3402 Administration Enhances Mitochondrial Levels and Function in WATs Along with an Increase in Body Temperature in HFD Mice

Since UCP1 is a key molecule in mitochondrial thermogenesis, the effect of NV-LJ3402 on mitochondrial content was determined in the WATs of HFD mice and 3T3-L1 adipocytes. NV-LJ3402 increased the mtDNA copy number in WAT (iWAT and eWAT) of HFD-fed mice (*p* < 0.05 vs. HFD, Figure 4a). Furthermore, HFD-NV-LJ3402 mice also showed enhanced citrate synthase activity (*p* < 0.05 vs. HFD), which is an indicator of mitochondrial function in WATs (Figure 4b). In addition, 48-h treatment of day 6 3T3-L1 adipocytes with NV-LJ3402-CM resulted in increased mtDNA copy number and citrate synthase activity (*p* < 0.05 vs. con), indicating that NV-LJ3402 can enhance the mitochondrial number and function in adipocytes (Figure 4c). Consistent with the NV-LJ3402-induced increase in mitochondrial number and function in the WAT of HFD-fed mice, HFD-NV-LJ3402 mice showed higher core body temperatures at 25 °C (0 h) and 4 °C (2‒6 h) than HFD mice. However, the effect of NV-LJ3402 on core body temperature was statistically meaningful only at 4–6 h cold exposure (*p* < 0.05). However, the effect of NV-LJ3402 on body temperature increase was significant (*p* < 0.05) after exposure to a temperature of 4 °C for 4 h (Figure 4d). Together, NV-LJ3402 increased the body temperature by enhancing thermogenesis by increased levels of mitochondria.

### 3.5. Applicability of NV-LJ3042

As shown in Table 2, the addition of NV-LJ3402 (10^8^ and 10^9^) did not affect the pH development and growth of the starter during yogurt fermentation.

The fermentation time to reach pH 4.5 was shown to be 4 h 31 min (Control), 4 h 27 min (NV-LJ3402, 10^8^), and 4 h 24 min (NV-LJ3402, 10^9^). With the addition of NV-LJ3402, the incubation time was a few minutes faster than that of the control. However, a few-minute difference does not hold significance in industrial applications.

After yogurt fermentation, the viable cells for *S. thermophilus* increased to 1.68 × 10^9^ (Control), 1.71 × 10^9^ (NV-LJ3402, 10^8^), and 1.37 × 10^9^ cfu/mL (NV-LJ3402, 10^9^), respectively. Whereas, *L. delbrueckii* subsp. *bulgaricus* increased to 9.17 × 10^8^ (Control), 9.26 × 10^8^ (NV-LJ3402, 10^8^), and 1.19 × 10^9^ cfu/mL (NV-LJ3402, 10^9^), respectively. However, there was no significant differences in the viable cells of *S. thermophilus* and *L.*
*delbrueckii* subsp. *bulgaricus*.

## 4. Discussion

Obesity is a major health problem in westernized societies because it is closely associated with various metabolic complications, including diabetes, hepatic steatosis, and cardiovascular diseases. Therefore, obesity prevention has important health benefits. Many studies have shown that several strains of probiotic bacteria could reduce insulin resistance and the incidence of diet-induced obesity, insulin resistance, diabetes, and fatty liver diseases [1]. Recent studies reported that two strains of *Lactobacillus* sp. exert a protective effect against diet-induced obesity by regulating hepatic fatty acid metabolism and inducing white adipose browning, respectively [3,4]. However, most studies investigating the effects of probiotics on host health, including these studies, have been performed using live bacteria. Although some reports have shown that NV probiotics play a role in the host immune system and reduce diet-induced obesity and insulin resistance [19,20,21], little is still known about the effect of NV probiotic bacteria on energy homeostasis in HFD-fed mice. Here, the effect of NV-LJ3402 on diet-induced obesity was examined, and the results demonstrated that NV-LJ3402 attenuates body weight gain and adiposity in HFD-fed mice by enhancing the expression of genes critical for energy dissipation in WAT.

Many studies have reported that chronic overnutrition causes alterations in the expression or activity of metabolic transcription factors or both, which results in the deregulation of the expression of their target genes, coding for proteins involved in energy metabolism, consequently leading to metabolic abnormalities such as obesity and diabetes [22]. Therefore, the administration of NV-LJ3402 may continuously attenuate HFD-induced deregulation of metabolic gene expression or activity, and this accumulated effect of NV-LJ3402 may protect against diet-induced obesity. Consistent with this assumption and the results of other studies [3,4,23,24], continuous administration of NV-LJ3402 resulted in amelioration of diet-induced obesity in the later stages of HFD feeding. In this study, NV-LJ3402 upregulated the expression of PPARγ and its target genes that are important for energy dissipation (ACOX and UCP1) in WATs and downregulated the expression of lipogenic genes. Furthermore, both NV-LJ3402 CM and live LJ3402-CM increased the transcriptional activity of PPARγ to a similar extent in vitro—when reporter genes harboring the promoters of various PPARγ target genes were used—showing that both NV-LJ3402 and live LJ3402 could regulate the transcriptional activity of PPARγ.

The inhibitory effect of NV-LJ3402 on HFD-induced deregulation of these metabolic genes suggests that NV-LJ3402 could also ameliorate the HFD-induced alterations in the metabolic parameters. NV-LJ3402 administration reduced TG levels in the plasma and metabolic tissues (liver and eWAT) in HFD mice. Additionally, NV-LJ3402 administration reduced glucose and insulin levels in the plasma and resulted in a slight improvement in glucose tolerance, indicating that NV-LJ3402 reduces lipid accumulation and insulin resistance in HFD mice. Furthermore, NV-LJ3402 administration increased the body temperature in HFD-fed mice. Since NV-LJ3402 also increased the mitochondria number in WAT and enhanced WAT expression of UCP1, which is known to release energy in the form of heat by uncoupling ATP production through proton leakage in mitochondria. These results suggest that the NV-LJ3402-induced decrease in TG levels and the reduction in obesity in HFD mice could be due to increased energy expenditure. In addition, NV-LJ3402-induced reduction of lipid accumulation in mature 3T3-L1 adipocytes was observed in parallel with an increase in mitochondrial content. Taken together, these findings demonstrated that NV probiotic bacteria, NV-LJ3402, could ameliorate metabolic disorders such as obesity, which are induced by HFD, and that this beneficial effect of NV-LJ3402 might promote the development of safe dairy products aimed at attenuating diet-induced obesity.

## Figures and Tables

**Figure 1 foods-09-01494-f001:**
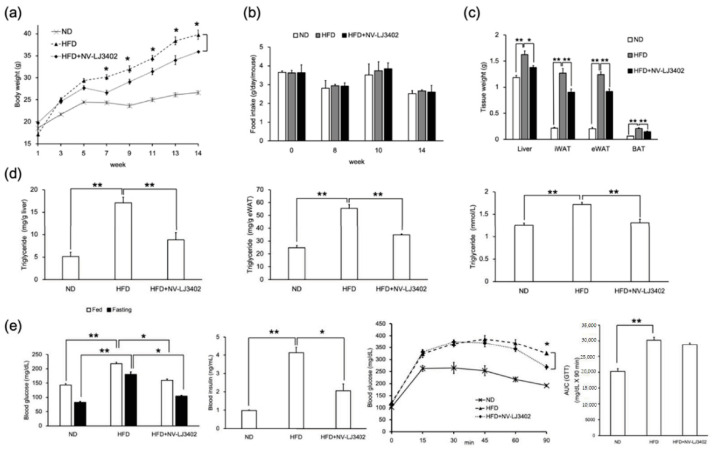
Non-viable *Lactobacillus johnsonii* JNU3402 (NV-LJ3402) attenuates diet-induced obesity in high fat diet (HFD)-fed mice. Seven-week-old C57BL/6J male mice were administered NV-LJ3402 or Phosphate-buffered saline (PBS) daily for 14 weeks during feeding with a normal diet (ND) or an HFD, and body weight gain (**a**) of mice was measured as indicated. (*n* = 7 per group). HFD vs. HFD+NV-LJ3402, * *p* < 0.05. Daily Food intake (**b**) was measured three times per indicated week, and tissue weights at the 14th week of feeding (**c**) were measured (B‒C, *n* = 6–7 per group). (**d**) Triglyceride (TG) levels in the tissues (liver and epididymal WAT (eWAT)) and plasma (*n* = 6–7 per group). (**e**) Plasma glucose, insulin levels, and glucose tolerance test results from each group of mice after 14 weeks of feeding (*n* = 7 per group). All data are expressed as the mean ± S.E.M. ND vs. HFD, HFD vs. HFD+NV-LJ3402, * *p* < 0.05, ** *p* < 0.01.

**Figure 2 foods-09-01494-f002:**
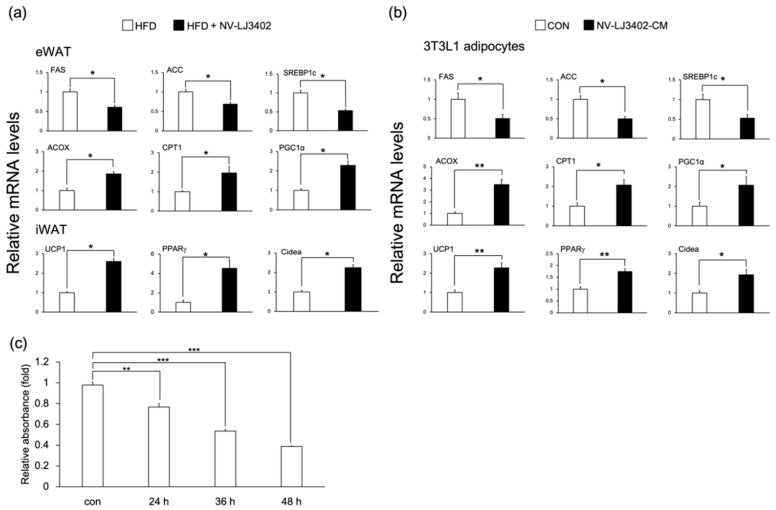
The effect of NV-LJ3402 on metabolic gene expression in the WATs of HFD-fed mice and 3T3-L1 adipocytes. After 14 weeks of NV-LJ3402 administration to HFD mice or 48-h treatment of day 6 3T3-L1 adipocytes with NV-LJ3402-CM or control bacterial culture medium (con), the expression of genes coding for proteins involved in lipogenesis and energy dissipation in WATs (**a**) and 3T3-L1 adipocytes (**b**) were determined by Real time-quantitative PCR (RT-qPCR), as indicated. (**c**) 3T3-L1 adipocytes were treated with NV-LJ3402-CM for different time periods, as indicated. Lipid accumulation in 3T3-L1 adipocytes was analyzed on day 8 by Oil Red O staining. All data are expressed as the mean ± S.E.M. The experiments using WATs were performed with *n* = 6–7 per group, and the results using 3T3-L1 adipocytes are from three independent experiments. HFD vs. HFD+NV-LJ3402, con vs. NV-LJ3402-CM, * *p* < 0.05, ** *p* < 0.01, *** *p* < 0.001.

**Figure 3 foods-09-01494-f003:**
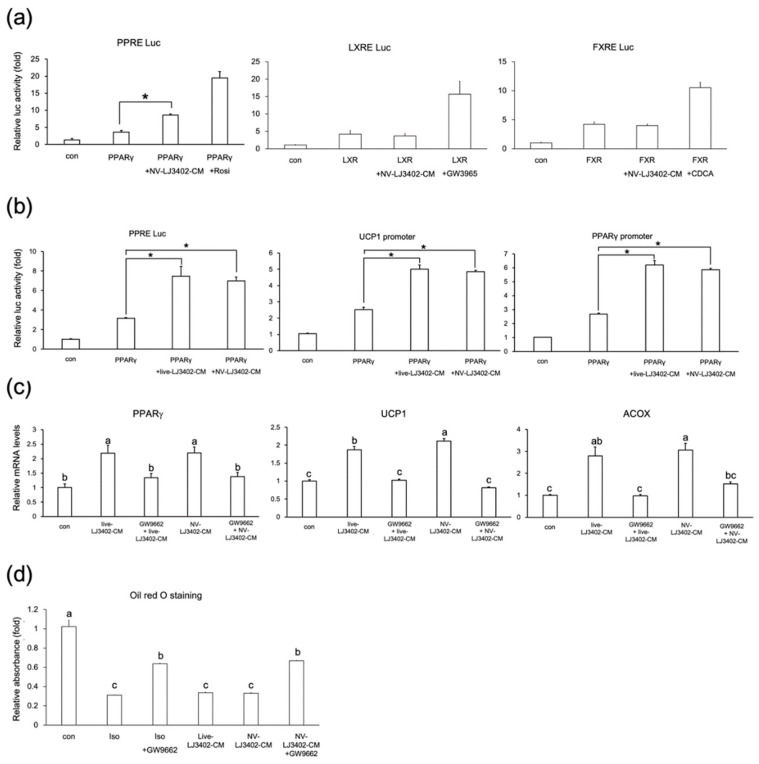
Effect of NV-LJ3402 on the activities of transcription factors important for metabolic gene expression (**a**) HEK293T cells were co-transfected with different metabolic transcription factors (PPARγ, liver X receptor, LXR or farnesoid X receptor, FXR) and appropriate reporter genes harboring their cognate response elements (pGL3-ACOX-PPRE-Luc, PPRE Luc, pGL3-CETP-LXRE-luc, LXRE Luc or pGL3-TK-IR-1-Luc, FXRE Luc), as indicated. After 12-h transfection, cells were incubated for 24 h with either NV-LJ3402-CM, appropriate ligands, or vehicles (DMSO or con or both), as indicated, and then assayed for luciferase activity. (**b**) Effect of LJ3402-CM on PPARγ transcriptional activity was analyzed using various reporter genes harboring the PPARγ response element (pGL3-ACOX-PPRE-Luc) or different target gene promoters (pGL3-UCP1-Luc and pGL3-PPARγ-Luc) in HEK293T cells. After 12-h transfection, cells were treated with con, live LJ3402-CM, and NV-LJ3402-CM as indicated. (**c**,**d**) Day 6 3T3-L1 adipocytes were incubated with live LJ3402-CM, NV-LJ3402-CM, isoproterenol (Iso, 100 μM) or GW9662 (10 μM) or both for 48 h, as indicated. The mRNA levels (**c**) of genes coding for proteins involved in energy dissipation and lipid accumulation (**d**) were analyzed by RT-qPCR and Oil Red O staining, respectively, as indicated. All data are expressed as the mean ± S.E.M of three independent experiments. PPARγ vs. PPARγ+NV-LJ3402CM or live LJ3402-CM, * *p* < 0.05. Different lower-case letters above bars indicate significant differences (*p* < 0.05).

**Figure 4 foods-09-01494-f004:**
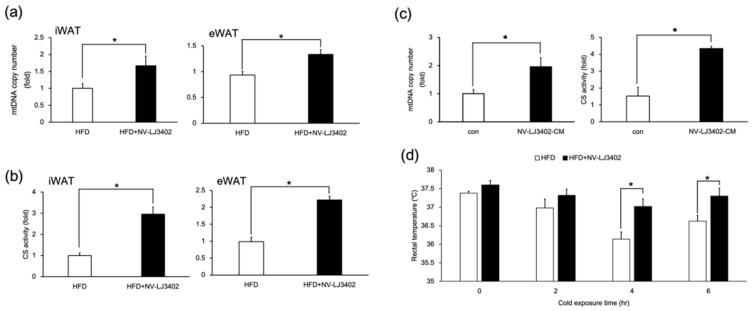
NV-LJ3402 enhances mitochondrial levels and increases the body temperature in HFD-fed mice. (**a**,**b**) Mitochondrial DNA (mtDNA) copy number (**a**) and citrate synthase activity (**b**) in inguinal WAT (iWAT) and eWAT of HFD and HFD-NV-LJ3402 mice (*n* = 5–7 per group). (**c**) Day 6 3T3-L1 adipocytes were treated with NV-LJ3402-CM for 48 h, and then mtDNA copy number and citrate synthase activity were measured on day 8, as indicated. mtDNA levels were normalized to those of nuclear genomic DNA. The results are from three independent experiments. (**d**) Core body temperature was measured based on rectal temperatures at 25 °C (0 h) and 4 °C for different durations (2‒6 h) (*n* = 7 per group). All data are expressed as the mean ± S.E.M. HFD vs. HFD+NV-LJ3402, con vs. NV-LJ3402-CM, * *p* < 0.05.

**Table 1 foods-09-01494-t001:** Primers used for the Real time-quantitative PCR (RT-qPCR).

Gene	5′—Sense Primer—3′	5′—Antisense Primer—3′
FAS	AGATCCTGGAACGAGAACACGAT	GAGACGTGTCACTCCTGGACTTG
Acc	GTATGTTCGAAGGGCTTACATTG	TGGGCAGCATGAACTGAAATT
SREBP1c	ACTGTGACCTCACAGGTCCA	GGCAGTTTGTCTGTGTCCACA
ACOX	TCGAGGCTTGGAAACCACTG	TCGAGTGATGAGCTGAGCC
CPT1	ACTCCTGGAAGAAGAAGTTC	TAGGGTCCGATTGATCTTTG
PGC1α	GAGACTTTGGAGGCCAGCA	CGCCATCCCTTAGTTCACTGG
UCP1	GGAGGTGTGGCAGTGTTC	TCTGTGGTGGCTATAACTCTG
PPARγ	GAAGACCACTCGCATTCCTT	GAAGGTTCTTCATGAGGCCTG
Cidea	ATCACAACTGGCCTGGTTACG	TACTACCCGGTGTCCATTTCT
36B4	AGATGCAGCAGATCCGCAT	ATATGAGGCAGCAGTTTCTCCAG
D-loop	AATCTACCATCCTCCGTG	GACTAATGATTCTTCACCGT
GAPDH	GTTGTCTCCTGCGACTTCA	GGTGGTCCAGGGTTTCTTA

**Table 2 foods-09-01494-t002:** Changes of the pH value and the viable cells of lactic acid bacteria in yogurt fermentation.

	Initial Fermentation	Final Fermentation
	(Means ± SD)	(Means ± SD)
Control	6.49	4.5 (4 h 31 min) *
pH	5.23 × 10^6^ ± 0.98	1.68 × 10^9^ ± 1.06
*S. thermophilus* *L. bulgaricus* subsp. *bulgaricus*	4.83 × 10^6^ ± 1.71	9.17 × 10^8^ ± 0.65
Non-viable *L. johnsonii* JNU3402 (10^8^) pH	6.52	4.5 (4 h 27 min) *
*S. thermophilus*	5.07 × 10^6^ ± 1.11	1.71 × 10^9^ ± 0.82
*L. bulgaricus* subsp. *bulgaricus*	4.65 × 10^6^ ± 1.62	9.26 × 10^8^ ± 2.95
Non-viable *L. johnsonii* JNU3402 (10^9^)	6.54	4.5 (4 h 24 min) *
pH	5.16 × 10^6^ ± 1.18	1.37 × 10^9^ ± 0.65
*S. thermophilus* *L. bulgaricus* subsp. *bulgaricus*	4.37 × 10^6^ ± 0.95	1.19 × 10^9^ ± 0.46

* Fermentation time to reach pH 4.5.

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
