# Peer review of "Non-Viable Lactobacillus johnsonii JNU3402 Protects against Diet-Induced Obesity"

_foods, 2020, doi:10.3390/foods9101494_

Round 1

Reviewer 1 Report

The paper by Yang et al. added information to the therapeutic potential of non-viable microorganisms as NV-LJ3402 in a preclinical model of diet-induced obesity.

The described topic is pertinent for the journal Foods in which the paper has been submitted. 

Although the research design appears appropriate, results and methods need to be improved in the following issues:

  1. Material & Methods section need to be better detailed. Specifically, the Authors should added the description of the preparation of the control bacteria medium (con) that they used in fig. 2b. In the statistical analysis paragraph, the Authors should describe the statistical test they used for the data analysis. Student t-test is applicable only in the case of two experimental conditions.
  2. The Authors should indicate in each figure caption the data representation (i.e., mean ± SD or SEM) and how many experiments the mean represents.
  3. In the weight gain curves represented in Fig.1a, it is not clear what the significant difference is referred to and which statistical analysis was applied.
  4. The title of the y-axis of fig. 1b, indicating the daily food intake, should be corrected in ‘g/day/mouse’.
  5. Please, detail what the significant difference in fig.1e (glucose tolerance test) is referred to. It should more appropriate to represent the AUC.

The Authors should better discuss the delayed effect (14th week) observed in the mice after an early treatment (the first 14 days) with NV-LJ3402.

Minor text editing

Page 1 line 29 eliminate ‘r’ from ‘promoter’.

Overall, the manuscript need a major revision mainly focused on the statistical analysis of the results.

Author Response

Reviewer #1

The paper by Yang et al. added information to the therapeutic potential of non-viable microorganisms as NV-LJ3402 in a preclinical model of diet-induced obesity.

The described topic is pertinent for the journal Foods in which the paper has been submitted. 

Although the research design appears appropriate, results and methods need to be improved in the following issues:

1-1. Material & Methods section need to be better detailed. Specifically, the Authors should added the description of the preparation of the control bacteria medium (con) that they used in fig. 2b.

Answer: We thank the reviewer for this comment. As with the NV-LJ3402-CM, the control bacterial culture medium (con) was prepared by heating the MRS broth at 80 °C for 15 min. As suggested by the reviewer, we have added this information to the “Materials and Methods” section.

Page 2, line 83:

2.3. Cell Culture and Transfection

HEK293T and 3T3-L1 cells were cultured in Dulbecco's Modified Eagle Medium containing 5% fetal bovine serum or 10% newborn calf serum and antibiotics. Plasmids, pGL3-UCP1 promoter (-2620 to +68 bp), pGL3-ACOX-PPRE-Luc, pGL3-CETP-LXRE-Luc, pGL3-TK-IR-1-Luc, pCDNA3-PPARg, pCMX-RXRa, pCMX-LXR, and pSG5-FXR, have been described previously [3,13-15]. MRS broth was heated at 80 °C for 15 min and used as a control bacterial culture medium (con). NV-LJ3402-CM or con was added to 3T3-L1 adipocytes on day 6 for 48 h at 1/100 volume of the medium to test the effect of NV-LJ3402-CM on gene expression, mitochondrial levels, and lipid accumulation. In addition, NV-LJ3402-CM was added to HEK293T cells for 24 h to test its effect on the activities of transcription factors. Adipocyte differentiation of 3T3-L1 cells and transfection were performed as previously described [4].

1-2. In the statistical analysis paragraph, the Authors should describe the statistical test they used for the data analysis. Student t-test is applicable only in the case of two experimental conditions.

Answer: We thank the reviewer for this pertinent comment. We have used the Student’s t-test (two-sample t-test) only for comparisons between two groups (i.e., ND vs. HFD, HFD vs. HFD-NV-LJ3402, con vs. NV-LJ3402-CM, and PPARg vs. PPARg+NV-LJ3402-CM). For comparisons between more than two groups, we have used Tukey’s multiple comparison test. We have described this statistical analysis in the “Materials and Methods” section.

Page 4, line 127:

2.9. Statistical Analysis

  1. The Authors should indicate in each figure caption the data representation (i.e., mean ± SD or SEM) and how many experiments the mean represents.

Answer: We thank the reviewer for this pertinent suggestion. Accordingly, we have included the data representation and the number of experiments in the figure legends.

Figure 1. NV-LJ3402 attenuates diet-induced obesity in HFD-fed mice. Seven-week-old C57BL/6J male mice were administered NV-LJ3402 or PBS daily for 14 weeks during feeding with a normal diet (ND) or a HFD, and body weight gain (a) of mice was measured as indicated (n = 7 per group). HFD vs. HFD+NV-LJ3402; *P < 0.05. Daily Food intake (b) was measured three times per indicated week, and tissue weights at the 14th week of feeding (c) were measured (B‒C; n = 6–7 per group). (d) Triglyceride (TG) levels in the tissues (liver and eWAT) and plasma (n =6–7 per group). (e) Plasma glucose, insulin levels, and glucose tolerance test results from each group of mice after 14 weeks of feeding (n = 7 per group). All data are expressed as the mean ± S.E.M. ND vs. HFD, HFD vs. HFD+NV-LJ3402; *P < 0.05, ** P < 0.01.

Figure 2. The effect of NV-LJ3402 on metabolic gene expression in the WATs of HFD-fed mice and 3T3-L1 adipocytes. After 14 weeks of NV-LJ3402 administration to HFD mice or 48-h treatment of day 6 3T3-L1 adipocytes with NV-LJ3402-CM or control bacterial culture medium (con), the expression of genes coding for proteins involved in lipogenesis and energy dissipation in WATs (a) and 3T3-L1 adipocytes (b) were determined by RT-qPCR, as indicated. (c) 3T3-L1 adipocytes were treated with NV-LJ3402-CM for different time periods, as indicated. Lipid accumulation in 3T3-L1 adipocytes was analyzed on day 8 by Oil Red O staining. All data are expressed as the mean ± S.E.M. The experiments using WATs were performed with n = 6–7 per group, and the results using 3T3-L1 adipocytes are from three independent experiments. HFD vs. HFD+NV-LJ3402, con vs. NV-LJ3402-CM; *P < 0.05, **P < 0.01, ***P < 0.001.

Figure 3. Effect of NV-LJ3402 on the activities of transcription factors important for metabolic gene expression (a) HEK293T cells were co-transfected with different metabolic transcription factors (PPARg, liver X receptor; LXR or farnesoid X receptor; FXR) and appropriate reporter genes harboring their cognate response elements (pGL3-ACOX-PPRE-Luc; PPRE Luc, pGL3-CETP-LXRE-luc; LXRE Luc or pGL3-TK-IR-1-Luc; FXRE Luc), as indicated. After 12-h transfection, cells were incubated for 24 h with either NV-LJ3402-CM, appropriate ligands, or vehicles (DMSO or con or both), as indicated, and then assayed for luciferase activity. (b) Effect of LJ3402-CM on PPARg transcriptional activity was analyzed using various reporter genes harboring the PPARg response element (pGL3-ACOX-PPRE-Luc) or different target gene promoters (pGL3-UCP1-Luc and pGL3-PPARg-Luc) in HEK293T cells. After 12-h transfection, cells were treated with con, live LJ3402-CM, and NV-LJ3402-CM as indicated. (c-d) Day 6 3T3-L1 adipocytes were incubated with live LJ3402-CM, NV-LJ3402-CM, isoproterenol (Iso; 100 mM) or GW9662 (10 mM) or both for 48 h, as indicated. The mRNA levels (c) of genes coding for proteins involved in energy dissipation and lipid accumulation (d) were analyzed by RT-qPCR and Oil Red O staining, respectively, as indicated. All data are expressed as the mean ± S.E.M of three independent experiments. PPARg vs. PPARg+NV-LJ3402CM or live LJ3402-CM; *P < 0.05. Different lower-case letters above bars indicate significant differences (P < 0.05).

Figure 4. NV-LJ3402 enhances mitochondrial levels and increases the body temperature in HFD-fed mice. (a-b) Mitochondrial DNA (mtDNA) copy number (a) and citrate synthase activity (b) in iWAT and eWAT of HFD and HFD-NV-LJ3402 mice (n = 5–7 per group). (c) Day 6 3T3-L1 adipocytes were treated with NV-LJ3402-CM for 48 h, and then mtDNA copy number and citrate synthase activity were measured on day 8, as indicated. mtDNA levels were normalized to those of nuclear genomic DNA. The results are from three independent experiments. (d) Core body temperature was measured based on rectal temperatures at 25 ℃ (0 h) and 4 ℃ for different durations (2‒6 h) (n = 7 per group). All data are expressed as the mean ± S.E.M. HFD vs. HFD+NV-LJ3402, con vs. NV-LJ3402-CM; *P < 0.05.

  1. In the weight gain curves represented in Fig.1a, it is not clear what the significant difference is referred to and which statistical analysis was applied.

Answer: We thank the reviewer for pointing this out. As described in the “Material and Methods” section, all values are expressed as mean ± S.E.M, and the statistical significance was analyzed by Student’s t-test for comparison between two groups. Thus, the values in Figure 1 are expressed as mean ± S.E.M, and the comparison of body weight gain between the HFD and HFD-NV-LJ3402 groups was assessed by Student’s t-test. However, to underline this more clearly, we rewrote our description in the “Results” section and figure legends.

Page 4, line 141:

Administration of NV-LJ3402 to HFD-fed mice (NV-LJ3402 mice) reduced HFD-induced body weight gain by 10% (HFD, 39.79 ± 1.04 g vs. NV-LJ3402, 35.95 ± 1.01 g; P < 0.05) in parallel with a reduction in liver weight (14%; HFD, 1.63 ± 0.07 g vs. NV-LJ3402, 1.37 g ± 0.04 g; P < 0.05), epididymal WAT (eWAT; 26%; HFD, 1.24 ± 0.05 g vs. NV-LJ3402, 0.92 g ± 0.05 g; P < 0.01), inguinal WAT (iWAT; 30%; HFD, 1.27 ± 0.09 g vs. NV-LJ3402, 0.90 g ± 0.06 g; P < 0.01), and brown adipose tissue weight (30%; HFD, 0.21 ± 0.01 g vs. NV-LJ3402, 0.15 g ± 0.01 g; P < 0.01).

Page 5, lines 173 and 178:

Figure 1. NV-LJ3402 attenuates diet-induced obesity in HFD-fed mice. Seven-week-old C57BL/6J male mice were administered NV-LJ3402 or PBS daily for 14 weeks during feeding with a normal diet (ND) or a HFD, and body weight gain (a) of mice was measured as indicated (n = 7 per group). HFD vs. HFD+NV-LJ3402; *P < 0.05. Daily Food intake (b) was measured three times per indicated week, and tissue weights at the 14th week of feeding (c) were measured (B‒C; n = 6–7 per group). (d) Triglyceride (TG) levels in the tissues (liver and eWAT) and plasma (n =6–7 per group). (e) Plasma glucose, insulin levels, and glucose tolerance test results from each group of mice after 14 weeks of feeding (n = 7 per group). All data are expressed as the mean ± S.E.M. ND vs. HFD, HFD vs. HFD+NV-LJ3402; *P < 0.05, ** P < 0.01.

  1. The title of the y-axis of fig. 1b, indicating the daily food intake, should be corrected in ‘g/day/mouse’.

Answer: We thank the reviewer for this pertinent suggestion. We have revised the title of the y-axis in Figure 1b as “Food intake (g/day/mouse)” and changed the corresponding graph accordingly.

  1. Please, detail what the significant difference in fig.1e (glucose tolerance test) is referred to. It should more appropriate to represent the AUC.

Answer: We thank the reviewer for this comment. With the statistical analyses of the glucose and insulin levels at all time points, we also performed AUC analysis for the glucose tolerance test (GTT). In the AUC analysis, we observed a mild reduction in glucose levels in NV-LJ3402 mice; however, this glucose-lowering effect was not statistically significant until 60 min after glucose injection (P = 0.0934 vs. HFD). However, 90 min after glucose injection, this effect became significant (P < 0.05 vs. HFD), which is indicated by an asterisk in Figure 1e (blood glucose level). This suggests that the administration of NV-LJ3402 mildly enhances insulin sensitivity in HFD mice.

To make this observation more clearly in the revised manuscript, we have rewritten the description of the effect of NV-LJ3402 on insulin sensitivity in the “Abstract”, “Results”, and “Discussion” sections.

Abstract (Page 1, line 16):

Concomitantly, NV -LJ3402 administration to HFD-fed mice also decreased the triglyceride levels in the plasma and metabolic tissues and slightly improved insulin resistance.

Results (Page 4, line 162):

As expected, intraperitoneal glucose injections increased plasma glucose levels in both HFD and NV-LJ3402 mice to a similar level (Figure 1e). The area under the curve (AUC) analysis for the glucose tolerance test showed that NV-LJ3402 mice exhibited slightly lower glucose levels than HFD mice; however, the glucose-lowering effect of NV-LJ3402 was not statistically significant (P = 0.0934 vs. HFD). NV-LJ3402 administration reduced plasma glucose levels after 45 min post glucose injection. This NV-LJ3402-mediated reduction in plasma glucose levels became significant 90 min after glucose injection, while the plasma glucose levels in HFD-fed mice did not significantly change until this time.

Discussion (Page 10, line 345):

Additionally, NV-LJ3402 administration reduced glucose and insulin levels in the plasma and resulted in a slight improvement in glucose tolerance, indicating that NV-LJ3402 reduces lipid accumulation and insulin resistance in HFD mice.

The Authors should better discuss the delayed effect (14th week) observed in the mice after an early treatment (the first 14 days) with NV-LJ3402.

Answer: We thank the reviewer for this pertinent comment. It is well-known that diet-induced obesity is caused by chronic overnutrition, leading to disturbances in lipid metabolism by the deregulation of metabolic genes. Dysregulation of lipid metabolism, in combination with an increased flux of dietary fatty acids, progressively promotes excessive fat accumulation in the adipose tissue, resulting in obesity and associated metabolic disorders after a certain time. Therefore, we believe that NV-LJ3402 continuously inhibits overnutrition-induced deregulation of metabolic genes that occur chronically, not acutely. This may contribute to the delayed onset of the NV-LJ3402-mediated effect on diet-induced obesity. Consistently, our and other studies on the effect of Lactobacillus strains on diet-induced obesity have shown that their anti-obesity effects only occur several weeks after their administration [3,4,23,24]

To explain the delayed effect of NV-LJ3402 on the protection of diet-induced obesity in the main text, we have added the following sentences in the “Discussion” section (page 10, line 331):

Many studies have reported that chronic overnutrition causes alterations in the expression or activity of metabolic transcription factors, or both, which results in the deregulation of the expression of their target genes, coding for proteins involved in energy metabolism, consequently leading to metabolic abnormalities such as obesity and diabetes [22]. Therefore, the administration of NV-LJ3402 may continuously attenuate HFD-induced deregulation of metabolic gene expression or activity, and this accumulated effect of NV-LJ3402 may protect against diet-induced obesity. Consistent with this assumption and the results of other studies [3,4,23,24], continuous administration of NV-LJ3402 resulted in amelioration of diet-induced obesity in the later stages of HFD feeding.

References

  1. Park, S. S.; Lee, Y. J.; Song, S.; Kim, B.; Kang, H.; Oh, S.; Kim, E. Lactobacillus acidophilus NS1 attenuates diet-induced obesity and fatty liver. J Endocrinol 2018, 237, 87-100, doi:10.1530/JOE-17-0592.
  2. Park , S.S.; Lee, Y.J.; Kang, H.; Yang, G.; Hong, E.J.; Lim, J.Y.; Oh, S.; Kim E. Lactobacillus amylovorus KU4 ameliorates diet-induced obesity in mice by promoting adipose browning through PPARgamma signaling. Sci Rep 2019, 9, 20152, doi:10.1038/s41598-019-56817-w.
  3. Cheng, Y.C.; Liu, J.R. Effect of LGG on energy metabolism, leptin resistance, and gut microbiota in mice with diet-induced obesity. Nutrients 2020, 12, doi:10.3390/nu12092557.
  4. Hsieh, M.C.; Tsai, W.H.; Jheng, Y.P.; Su, S.L.; Wang, S.Y.; Lin, C.C.; Chen, Y.H.; Chang, W.W. The beneficial effects of Lactobacillus reuteri ADR-1 or ADR-3 consumption on type 2 diabetes mellitus: a randomized, double-blinded, placebo-controlled trial. Sci Rep 2018, 8, 16791, doi:10.1038/s41598-018-35014-1.

Minor text editing

Page 1 line 29 eliminate ‘r’ from ‘promoter’.

Answer: We thank the reviewer for spotting this error; we have corrected this spelling error.

Overall, the manuscript need a major revision mainly focused on the statistical analysis of the results.

Answer: We thank the reviewer for this suggestion. We have revised the text associated with the statistical analysis in the “Results” section. In addition, statistical differences in gene expression and lipid accumulation among different treatment groups were analyzed by Tukey’s multiple comparison test (please see the “Materials and Methods” section in the revised manuscript). Accordingly, we replaced our data in Figure 3c and d (figures below).

Results section

Page 4, line 141:

Administration of NV-LJ3402 to HFD-fed mice (NV-LJ3402 mice) reduced HFD-induced body weight gain by 10% (HFD, 39.79 ± 1.04 g vs. HFD+NV-LJ3402, 35.95 ± 1.01 g; P < 0.05) in parallel with a reduction in liver weight (14%; HFD, 1.63 ± 0.07 g vs. HFD+NV-LJ3402, 1.37 g ± 0.04 g; P < 0.05), epididymal WAT (eWAT, 26%; HFD, 1.24 ± 0.05 g vs. HFD+NV-LJ3402, 0.92 g ± 0.05 g; P < 0.01), inguinal WAT (iWAT, 30%; HFD, 1.27 ± 0.09 g vs. HFD+NV-LJ3402, 0.90 g ± 0.06 g; P < 0.01), and brown adipose tissue weight (30%; HFD, 0.21 ± 0.01 g vs. HFD+NV-LJ3402, 0.15 g ± 0.01 g; P < 0.01).

Page 4, line 148:

Furthermore, TG levels in eWAT and the liver in NV-LJ3402 mice were reduced by 36% (34.87 ± 0.74 mg/g in eWAT; P < 0.01) and 70% (8.84 ± 1.60 mg/g in the liver; P < 0.01), respectively, as compared with those (55.43 ± 3.01 mg/g in eWAT and 17.09 ± 1.27 mg/g in the liver) in HFD-fed mice (HFD mice) (Figure 1d).

Page 4, line 153:

However, when NV-LJ3402 was administered during HFD feeding, plasma TG levels in HFD mice were reduced by 20% (HFD, 1.71 ± 0.05 mmol/L vs. HFD+NV-LJ3402, 1.33 ± 0.07 mmol/L; P < 0.01), indicating that the NV-LJ3402 administration confers resistance to HFD-induced obesity and lipid accumulation in the metabolic tissues.

Page 4, lines 157 and 159:

As expected, HFD increased the postprandial and fasting plasma glucose levels by 1.5- and 2-fold, respectively, compared to those in ND-fed mice (P < 0.01, Figure 1e). However, when NV-LJ3402 was administered to HFD mice, fed and fasting plasma glucose levels and plasma insulin levels were reduced to 26% (P < 0.05), 40% (P < 0.05), and 51% (P < 0.05), respectively, compared to those in HFD mice.

Page 5, line 185 and 192 – Page 6, line 194:

NV-LJ3402 reduced the mRNA levels of lipogenic genes, such as fatty acid synthase (FAS), acetyl-CoA carboxylase (ACC), and sterol regulatory element-binding protein-1c (SREBP1c), compared with those in eWAT of HFD mice (P < 0.05). Conversely, NV-LJ3402 increased the mRNA levels of genes coding for proteins involved in beta-oxidation, such as acetyl-CoA oxidase (ACOX), carnitine palmitoyltransferase 1 (CPT1), and peroxisome proliferator-activated receptor gamma coactivator 1-α (PGC1α) in eWAT of HFD mice (P < 0.05 vs. HFD, Figure 2a). Furthermore, NV-LJ3402 mice also exhibited increased mRNA levels of browning genes in iWAT (P < 0.05), such as uncoupling protein 1 (UCP1), PPARg, and Cidea, compared with those in eWAT of HFD mice. Additionally, when day 6 3T3-L1 adipocytes (day 6 after differentiation) were treated with NV-LJ3402-CM for 48 h, the mRNA levels of genes coding for proteins regulating energy dissipation (UCP1, PPARg, Cidea, ACOX, CPT1, and PGC1α) were significantly increased (P < 0.05 or P < 0.01 vs. con) in parallel with the reduction in mRNA levels of lipogenic genes (FAS, ACC, and SREBP1c; P < 0.05 vs. con, Figure 2b).

Page 6, line 204:

When 3T3-L1 adipocytes were treated with NV-LJ3402-CM, lipid accumulation on day 8 progressively reduced depending on the treatment duration (22%, 46%, and 61% decrease after 24, 36, and 48 h treatment, respectively), compared to that in the control adipocytes (P < 0.01 or P < 0.001, Figure 2c).

Page 7, line 231:

As shown in Figure 3b, NV-LJ3402-CM increased PPARg activity by approximately 2.0‒2.3-fold, depending on the reporter genes, relative to PPARg alone (P < 0.05).

Page 7, line 233:

Furthermore, RT-qPCR showed that the mRNA levels of PPARg and PPARg target genes (ACOX and UCP1) were increased in day 8 3T3-L1 adipocytes treated with NV or live LJ3402-CM for 48 h (P < 0.05 vs. con, Figure 3c).

Page 7, line 247:

As expected, 48-h treatment of day 6 3T3-L1 adipocytes with a browning stimulus (100 mM isoproterenol), live LJ3402-CM, and NV-LJ3402-CM reduced lipid accumulation by 70%, 67%, and 68%, respectively, compared with that in the control adipocytes (P < 0.05, Figure 3d).

Page 8, lines 275, 278, and 282:

NV-LJ3402 increased the mtDNA copy number in WAT (iWAT and eWAT) of HFD-fed mice (P < 0.05 vs. HFD, Figure 4a). Furthermore, HFD-NV-LJ3402 mice also showed enhanced citrate synthase activity (P < 0.05 vs. HFD), an indicator of mitochondrial function in WATs (Figure 4b). In addition, 48-h treatment of day 6 3T3-L1 adipocytes with NV-LJ3402-CM resulted in increased mtDNA copy number and citrate synthase activity (P < 0.05 vs. con), indicating that NV-LJ3402 can enhance mitochondrial number and function in adipocytes (Figure 4c). Consistent with the NV-LJ3402-induced increase in mitochondrial number and function in the WAT of HFD-fed mice, HFD-NV-LJ3402 mice showed higher core body temperatures at 25 oC (0 h) and 4 oC (2‒6 h) than HFD mice. However, the effect of NV-LJ3402 on core body temperature was statistically meaningful only at 4-6 h cold exposure (P < 0.05).

Reviewer 2 Report

Review of:

Non-viable Lactobacillus johnsonii JNU3402 protects against diet-induced obesity

This nicely done study contains interesting data for dairy industry to develop new products containing non-viable bacteria presenting a positive effect on health. The manuscript is globally clear and needs some clarifications as mentioned below to try to improve it.   

The use of “we” or “our” should be avoided along the manuscript.

Line 23: I wouldn’t use the term “probiotic” as according to the current definition from WHO it must still be an alive bacteria.

Line 69: …..resuspended to reach a density of….

Line 70: The sentence should be clarified because it is not clear what mean “respectively” here. Was the aim to obtain 2 different concentrations? Maybe just remove the word “respectively”?

Line 87: Could the authors clarify what will be the use of NV-LJ3402-conditioned medium?

Line 106: sentence on quantity of mRNA to reformulate.

Line 110: “manufactured using” instead of “manufactured by”

Line 112: Could the authors explain why the NV fraction has been added before the fermentation process and not after?

Line 110-114: Could the authors add more explanations on the fermentation process: volume of fermentation, automatization of the process and pH measurements?

Line 128: The authors could cite the work of Cani. Many publications are of interest but here is an example:

Depommier C, Everard A, Druart C, Plovier H, Van Hul M, Vieira-Silva S, Falony G, Raes J, Maiter D, Delzenne NM, de Barsy M, Loumaye A, Hermans MP, Thissen JP, de Vos WM, Cani PD. Supplementation with Akkermansia muciniphila in overweight and obese human volunteers: a proof-of-concept exploratory study. Nat Med. 2019 Jul;25(7):1096-1103. doi: 10.1038/s41591-019-0495-2. Epub 2019 Jul 1. PMID: 31263284; PMCID: PMC6699990.

Results

3.1: Effect of NV-LJ3402 on HFD-induced body weight gain and adiposity in mice

Did the authors consider to have a group with ND + NV-LJ3402?

Line 150: Does the authors have additional data on plasma glucose level after 90 minutes? As the maximum of activity with the NV-LJ3402 seems to be at 90 minutes, it would be interesting to see the evolution at longer term.

Figure 1:

No standard deviation for the ND group in Fig.1a?

Fig. 1e: No “ND” group?

Line 198: “factors” instead of “fctors”

Line 271: Could the authors explain what the “N” is?

Discussion:

Line 290: others nicely done studies were performed with non viable bacteria. The authors should cite the studies of Patrice Cani’s team.

For example:

Plovier H, Everard A, Druart C, Depommier C, Van Hul M, Geurts L, Chilloux J, Ottman N, Duparc T, Lichtenstein L, Myridakis A, Delzenne NM, Klievink J, Bhattacharjee A, van der Ark KC, Aalvink S, Martinez LO, Dumas ME, Maiter D, Loumaye A, Hermans MP, Thissen JP, Belzer C, de Vos WM, Cani PD. A purified membrane protein from Akkermansia muciniphila or the pasteurized bacterium improves metabolism in obese and diabetic mice. Nat Med. 2017 Jan;23(1):107-113. doi: 10.1038/nm.4236. Epub 2016 Nov 28. PMID: 27892954.

Author Response

Reviewer #2

Non-viable Lactobacillus johnsonii JNU3402 protects against diet-induced obesity

This nicely done study contains interesting data for dairy industry to develop new products containing non-viable bacteria presenting a positive effect on health. The manuscript is globally clear and needs some clarifications as mentioned below to try to improve it.   

The use of “we” or “our” should be avoided along the manuscript.

Answer: We thank the reviewer for this pertinent suggestion. Accordingly, we have revised all sentences comprising the words “we” or “our” in the manuscript.

Page 1, line 9:

In this study, the role of non-viable Lactobacillus johnsonii JNU3402 (NV-LJ3402) in diet-induced obesity was investigated in mice fed a high-fat diet (HFD).

Page 1, line 22:

Together, these results suggest that NV-LJ3402 could be safely used to develop dairy products that ameliorate diet-induced obesity and hyperlipidemia.

Page 1, line 35:

Recent studies have shown that some Lactobacillus strains protect against diet-induced obesity in mice.

Page 2, lines 59:

Therefore, the effect of NV-LJ3402 on diet-induced obesity was determined using HFD-fed mice. This study demonstrated that NV-LJ3402 enhanced the expression of the metabolic genes involved in energy expenditure, at least in part by stimulating the proliferator-associated receptor-g (PPARg) activity and mitochondrial levels in WAT, thereby increasing the body temperature, resulting in protection from diet-induced obesity.

Page 2, line 66:

NV -LJ3402 was prepared using the following method.

Page 4, lines 155 and 160:

As diet-induced obesity is closely associated with insulin resistance, the possible effect of NV-LJ3402 on insulin resistance was determined. As expected, HFD increased the postprandial and fasting plasma glucose levels by 1.5- and 2-fold, respectively, compared to those in ND-fed mice (P < 0.01, Figure 1e). However, when NV-LJ3402 was administered to HFD mice, fed and fasting plasma glucose levels and plasma insulin levels were reduced to 26% (P < 0.05), 40% (P < 0.05), and 51% (P < 0.05), respectively, compared to those in HFD mice. Next, glucose tolerance was compared between HFD and NV-LJ3402 mice. As expected, intraperitoneal glucose injections increased plasma glucose levels in both HFD and NV-LJ3402 mice to a similar level (Figure 1e).

Page 5, line 185:

Therefore, to identify the metabolic pathway underlying the protective effect of NV-LJ3402 against HFD-induced obesity, RT-qPCR was performed to determine the expression profile of genes coding for proteins involved in lipid metabolism in the WAT.

Page 7, line 223:

To determine whether NV-LJ3402 regulates the expression of these metabolic genes by controlling the activity of transcription factors that play a key role in adipocyte biology, reporter gene assays were performed in HEK293T cells using reporter genes harboring binding sites of these transcription factors.

Page 7, line 229:

Next, to further confirm the induction of PPARg transcriptional activity by NV-LJ3402, two other reporter genes harboring the promoter element of either PPARg or UCP1 were used.

Page 7, line 244:

As the NV-LJ3402-induced increase in UCP1 and Cidea expression by enhancing PPARg activity was already observed in the subcutaneous iWAT of HFD mice and 3T3-L1 adipocytes, the potential inhibitory effect of GW9662 on the NV-LJ3402-induced reduction in lipid accumulation in 3T3-L1 adipocytes was determined using Oil Red O staining.

Page 9, line 274:

As UCP1 is a key molecule in mitochondrial thermogenesis, the effect of NV-LJ3402 on mitochondrial content was determined in the WATs of HFD mice and 3T3-L1 adipocytes.

Page 10, lines 318, 320 and 324:

Recent studies reported that two strains of Lactobacillus sp. exert a protective effect against diet-induced obesity by regulating hepatic fatty acid metabolism and inducing white adipose browning, respectively [3,4]. However, most studies investigating the effects of probiotics on host health, including these studies, have been performed using live bacteria. Although some reports have shown that NV probiotics play a role in the host immune system and reduce diet-induced obesity and insulin resistance [19-21], little is still known about the effect of NV probiotic bacteria on energy homeostasis in HFD-fed mice. Here, the effect of NV-LJ3402 on diet-induced obesity was examined, and the results demonstrated that NV-LJ3402 attenuates body weight gain and adiposity in HFD-fed mice by enhancing the expression of genes critical for energy dissipation in WAT.

Page 11, line 344:

The inhibitory effect of NV-LJ3402 on HFD-induced deregulation of these metabolic genes suggests that NV-LJ3402 could also ameliorate the HFD-induced alterations in the metabolic parameters. Indeed, NV-LJ3402 administration reduced TG levels in the plasma and metabolic tissues (liver and eWAT) in HFD mice.

Page 11, line 351, 352, and 354:

Furthermore, NV-LJ3402 administration increased the body temperature in HFD-fed mice. As NV-LJ3402 also increased the mitochondria number in WAT and enhanced WAT expression of UCP1, which is known to release energy in the form of heat by uncoupling ATP production through proton leakage in mitochondria. These results suggest that the NV-LJ3402-induced decrease in TG levels and the reduction in obesity in HFD mice could be due to increased energy expenditure. In addition, NV-LJ3402-induced reduction of lipid accumulation in mature 3T3-L1 adipocytes was observed in parallel with an increase in mitochondrial content. Taken together, these findings demonstrated that NV probiotic bacteria, NV-LJ3402, could ameliorate metabolic disorders such as obesity, which are induced by HFD, and that this beneficial effect of NV-LJ3402 might promote the development of safe dairy products aimed at attenuating diet-induced obesity.

Line 23: I wouldn’t use the term “probiotic” as according to the current definition from WHO it must still be an alive bacteria.

Answer: We thank the reviewer for pointing this out. As suggested by the reviewer, we have removed the term “probiotic” from the sentence.

Page 1, line 22:

Together, these results suggest that NV-LJ3402 could be safely used to develop dairy products that ameliorate diet-induced obesity and hyperlipidemia.

Line 69: …..resuspended to reach a density of….

Answer: We thank the reviewer for this comment and have revised the sentence accordingly.

Page 2, line 67:

After that, the pellets were then washed 3 times with sterile PBS (0.01 M, pH 7.2) and resuspended to reach a density of ca 1 × 108 or 1 × 109 cfu/mL.

Line 70: The sentence should be clarified because it is not clear what mean “respectively” here. Was the aim to obtain 2 different concentrations? Maybe just remove the word “respectively”?

Answer: We thank the reviewer for spotting this error. We have corrected this mistake accordingly.

Page 2, line 67:

After that, the pellets were then washed 3 times with sterile PBS (0.01 M, pH 7.2) and resuspended to reach a density of ca 1 × 108 or 1 × 109 cfu/mL.

Line 87: Could the authors clarify what will be the use of NV-LJ3402-conditioned medium?

Answer: We thank the reviewer for this pertinent comment. Accordingly, we have added detailed information about the use of the NV-LJ3402-conditioned medium to the “Materials and Methods” section.

Page 2, line 83 and 84:

2.3. Cell Culture and Transfection

HEK293T and 3T3-L1 cells were cultured in Dulbecco's Modified Eagle Medium containing 5% fetal bovine serum or 10% newborn calf serum and antibiotics. Plasmids, pGL3-UCP1 promoter (-2620 to +68 bp), pGL3-ACOX-PPRE-Luc, pGL3-CETP-LXRE-Luc, pGL3-TK-IR-1-Luc, pCDNA3-PPARg, pCMX-RXRa, pCMX-LXR, and pSG5-FXR, have been described previously [3,13-15]. MRS broth was heated at 80 °C for 15 min and used as a control bacterial culture medium (con). NV-LJ3402-CM or con was added to 3T3-L1 adipocytes on day 6 for 48 h at 1/100 volume of the medium to test the effect of NV-LJ3402-CM on gene expression, mitochondrial levels, and lipid accumulation. In addition, NV-LJ3402-CM was added to HEK293T cells for 24 h to test its effect on the activities of transcription factors. Adipocyte differentiation of 3T3-L1 cells and transfection were performed as previously described [4].

Line 106: sentence on quantity of mRNA to reformulate.

Answer: We thank the reviewer for this comment. To address this comment, we have added a description of the quantification by RT-qPCR to the “Materials and Methods” section.

Page 3, line 104:

2.6. RNA Isolation, Reverse Transcription, and RT-qPCR

Total RNA was isolated from 3T3-L1 cells and WATs using Trizol reagent (Invitrogen, Waltham, Massachusetts, USA), and cDNA was synthesized from 1 μg total RNA using M-MLV Reverse Transcriptase (Promega, Madison, USA). RT-qPCR was performed as previously described [4], and the results were normalized to 36B4 mRNA expression. Relative quantification of PCR products was calculated by the difference in Ct values between the target and 36B4 genes using the 2-ΔΔCt method. Primer sequences for PCR are listed in Table 1.

Line 110: “manufactured using” instead of “manufactured by”

Answer: We thank the reviewer for this suggestion. During the revision, we have rewritten the yogurt fermentation subsection in the “Materials and Methods” section and thereby removed the sentence mentioned by the reviewer. The revised yogurt fermentation subsection reads as follows (page 3, line 112):

2.7. Yogurt Fermentation

Homogenized whole milk (3.4% fat, 8.5% milk solid-non-fat; SNF) and skim milk powder (0.1% fat, 95% SNF) were obtained from a local dairy plant (Seoul Dairies, Seoul, Korea). Skim milk powder was added at a level of 2.5% to increase milk solids (11% SNF). One liter of milk base was heated at 95°C in a glass bottle (Schott Duran, Germany) for 10 min and then cooled to 42°C. Yogurt starters were inoculated with Streptococcus thermophilus (ca. 5 × 106 cfu/mL; Chr. Hansen Holdings A/S, Hoersholm, Denmark) and L. delbruekii subsp. bulgaricus (ca. 5 × 106 cfu/mL; Chr. Hansen Holdings A/S). After that, NV-LJ3402 (108 and 109) was added to the experimental group and then fermented at 42°C until the pH reached 4.5 [16].

Line 112: Could the authors explain why the NV fraction has been added before the fermentation process and not after?

Answer: We thank the reviewer for this comment. We are planning to produce a new type of yogurt commercially in collaboration with a dairy company in Korea. For this reason, we provided the company with non-viable LJ3402, and the company performed the fermentation. After solidifying milk (yogurt), we were not able to mix it to homogeneity. In addition, in the manufacturing process, opening the tank lid and adding the NV fraction after the fermentation process may introduce secondary contamination (from outside), leading to severe issues for the dairy company.

Line 110-114: Could the authors add more explanations on the fermentation process: volume of fermentation, automatization of the process and pH measurements?

Answer: We thank the reviewer for this pertinent comment. We used 1 L of standardized milk base (11% SNF), and the pH value was continuously recorded every 60 s during the fermentation process.

To share this additional information with the reader, as suggested by the reviewer, we have revised the text associated with yogurt fermentation. The revised text in the “Materials and Methods” section reads as follows:

Page 3, line 112:

2.7. Yogurt Fermentation

Homogenized whole milk (3.4% fat, 8.5% milk solid-non-fat; SNF) and skim milk powder (0.1% fat, 95% SNF) were obtained from a local dairy plant (Seoul Dairies, Seoul, Korea). Skim milk powder was added at a level of 2.5% to increase milk solids (11% SNF). One liter of milk base was heated at 95°C in a glass bottle (Schott Duran, Germany) for 10 min and then cooled to 42°C. Yogurt starters were inoculated with Streptococcus thermophilus (ca. 5 × 106 cfu/mL; Chr. Hansen Holdings A/S, Hoersholm, Denmark) and L. delbruekii subsp. bulgaricus (ca. 5 × 106 cfu/mL; Chr. Hansen Holdings A/S). After that, NV-LJ3402 (108 and 109) was added to the experimental group and then fermented at 42°C until the pH reached 4.5 [16].

Page 2, line 121:

2.8. Viable Cells and pH Measurements

The pH values were monitored using a Multi-Channel pH/Ion meter with a temperature probe (PhysioLab, Pusan, Korea) and recorded every 60 s during the fermentation process. Viable cells of L. delbrueckii subsp. bulgaricus were enumerated using MRS agar (BD, Difco Laboratories, Detroit, USA) adjusted to pH 5.4 and incubated anaerobically at 37°C for 48 h. For enumeration of S. thermophilus, diluted samples were incubated at 43°C for 24 h using M17 agar (BD).

Line 128: The authors could cite the work of Cani. Many publications are of interest but here is an example:

Depommier C, Everard A, Druart C, Plovier H, Van Hul M, Vieira-Silva S, Falony G, Raes J, Maiter D, Delzenne NM, de Barsy M, Loumaye A, Hermans MP, Thissen JP, de Vos WM, Cani PD. Supplementation with Akkermansia muciniphila in overweight and obese human volunteers: a proof-of-concept exploratory study. Nat Med. 2019 Jul;25(7):1096-1103. doi: 10.1038/s41591-019-0495-2. Epub 2019 Jul 1. PMID: 31263284; PMCID: PMC6699990.

Answer: We thank the reviewer for this suggestion. Recent studies, including the study mentioned by the reviewer, have shown that the administration of various probiotic bacteria improves obesity and associated metabolic disorders. We believe that the reviewer's reference is important for the present study to support the effect of probiotic bacteria on obesity and associated metabolic disorders because the study was conducted in human volunteers. Therefore, we have included this reference in the “Results” section.

Page 4, line 136:

Recent studies in mice and human volunteers showed that several strains of probiotic bacteria, such as Lactobacillus acidophilus NS1, Lactobacillus amylovorus KU4, and Akkermansia muciniphila, ameliorate obesity and associated metabolic disorders [3,4,17].

Reference

  1. Depommier, C., Everard, A., Druart, C., Plovier, H., Van Hul, M., Vieira-Silva, S., Falony, G., Raes, J., Maiter, D., Delzenne, N.M., et al. Supplementation with Akkermansia muciniphila in overweight and obese human volunteers: a proof-of-concept exploratory study. Nat Med 2019, 25, 1096-1103, doi:10.1038/s41591-019-0495-2.

Results

3.1: Effect of NV-LJ3402 on HFD-induced body weight gain and adiposity in mice

Did the authors consider to have a group with ND + NV-LJ3402?

Answer: We thank the reviewer for this pertinent comment. We agree that the administration of NV-LJ3402 to the ND group could provide valuable information on the role of NV-LJ3402 in energy metabolism under normal physiological conditions. However, this study focused on the protective effect of NV-LJ3402 on diet-induced obesity; therefore, we administered NV-LJ3402 only to HFD mice and determined its effect on reducing body weight gain and adiposity in HFD mice. We believe that future studies on the administration of NV-LJ3402 to the ND group will help us to gain the insight into the systemic role of NV-LJ3402 in energy homeostasis under normal physiological conditions.

Line 150: Does the authors have additional data on plasma glucose level after 90 minutes? As the maximum of activity with the NV-LJ3402 seems to be at 90 minutes, it would be interesting to see the evolution at longer term.

Answer: We thank the reviewer for pointing this out. We agree that extended time points in the GTT may further show the effect of NV-LJ3402 on HFD-induced insulin resistance. However, many studies (references 1–4, see below) measured glucose levels (using GTT) only until 60 or 90 min. because glucose measurements after longer periods could reflect complex physiology that goes beyond ‘insulin sensitivity’ Therefore, to avoid complications in the interpretation of the data, we performed GTT only for 90 min after glucose injection

References

  1. Park, S.S.; Lee, Y.J.; Kang, H.; Yang, G.; Hong, E.J.; Lim, J.Y.; Oh, S.; Kim, E. Lactobacillus amylovorus KU4 ameliorates diet-induced obesity in mice by promoting adipose browning through PPARgamma signaling. Sci Rep 2019, 9, 20152, doi:10.1038/s41598-019-56817-w.
  2. Park, S.S.; Lee, Y.J.; Song, S.; Kim, B.; Kang, H.; Oh, S.; Kim, E. Lactobacillus acidophilus NS1 attenuates diet-induced obesity and fatty liver. J Endocrinol 2018, 237, 87-100, doi:10.1530/JOE-17-0592.

3.Du, J.; Shen, L.; Tan, Z.; Zhang, P.; Zhao, X.; Xu, Y.; Gan, M.; Yang, Q.; Ma, J.; Jiang, A., et al. Betaine supplementation enhances lipid metabolism and improves insulin resistance in mice fed a high-fat diet. Nutrients 2018, 10, doi:10.3390/nu10020131.

  1. Gu, M.; Zhang, Y.; Fan, S.; Ding, X.; Ji, G.; Huang, C. Extracts of Rhizoma polygonati odorati prevent high-fat diet-induced metabolic disorders in C57BL/6 mice. PLoS One 2013, 8, e81724, doi:10.1371/journal.pone.0081724.

Figure 1:

No standard deviation for the ND group in Fig.1a?

Answer: We thank the reviewer for this comment. As described in the “Materials and Methods” section, all values are expressed as mean ± S.E.M. Thus, the data of the ND group shown in Figure 1a also include standard errors; however, these errors are rather small and consequently were concealed by data symbols (●). To avoid this concealing of standard errors in Figure 1a, we have replaced the old symbol (●) with the new symbol (x).

Fig. 1e: No “ND” group?

Answer: We thank the reviewer for this comment. Accordingly, we have replaced the old data in Figure 1e with new data containing plasma glucose and insulin levels of the ND group.

Line 198: “factors” instead of “fctors”

Answer: We thank the reviewer for spotting this error and have corrected it accordingly.

Page 6, line 223:

To determine whether NV-LJ3402 regulates the expression of these metabolic genes by controlling the activity of transcription factors that play a key role in adipocyte biology, reporter gene assays were performed in HEK293T cells using reporter genes harboring binding sites of these transcription factors.

  1. Line 271: Could the authors explain what the “N” is?

Answer: We thank the reviewer for pointing this out. This was an error, which we have corrected as follows (page 9, line 302):

With the addition of NV-LJ3402, the incubation time was a few minutes faster than that of the control.

Discussion:

Line 290: others nicely done studies were performed with non viable bacteria. The authors should cite the studies of Patrice Cani’s team.

For example:

Plovier H, Everard A, Druart C, Depommier C, Van Hul M, Geurts L, Chilloux J, Ottman N, Duparc T, Lichtenstein L, Myridakis A, Delzenne NM, Klievink J, Bhattacharjee A, van der Ark KC, Aalvink S, Martinez LO, Dumas ME, Maiter D, Loumaye A, Hermans MP, Thissen JP, Belzer C, de Vos WM, Cani PD. A purified membrane protein from Akkermansia muciniphila or the pasteurized bacterium improves metabolism in obese and diabetic mice. Nat Med. 2017 Jan;23(1):107-113. doi: 10.1038/nm.4236. Epub 2016 Nov 28. PMID: 27892954.

Answer: We thank the reviewer for this pertinent suggestion. We agree that the suggested reference, which describes the role of non-viable bacteria in energy homeostasis, fits with our study. As Akkermansia muciniphila is one of the most abundant bacterial species residing in the human intestine with probiotic properties, we have added the above reference to the “Discussion” section.

Page 10, line 322:

Recent studies reported that two strains of Lactobacillus sp. exert a protective effect against diet-induced obesity by regulating hepatic fatty acid metabolism and inducing white adipose browning, respectively [3,4]. However, most studies investigating the effects of probiotics on host health, including these studies, have been performed using live bacteria. Although some reports have shown that NV probiotics play a role in the host immune system and reduce diet-induced obesity and insulin resistance [19-21], little is still known about the effect of NV probiotic bacteria on energy homeostasis in HFD-fed mice. Here, the effect of NV-LJ3402 on diet-induced obesity was examined, and the results demonstrated that NV-LJ3402 attenuates body weight gain and adiposity in HFD-fed mice by enhancing the expression of genes critical for energy dissipation in WAT.

References

  1. Cani, P.D.; de Vos, W.M. Next-Generation Beneficial Microbes: The Case of Akkermansia muciniphila. Front Microbiol 2017, 8, 1765, doi:10.3389/fmicb.2017.01765.
  2. Depommier, C.; Van Hul, M.; Everard, A.; Delzenne, N.M.; De Vos, W.M.; Cani, P.D. Pasteurized Akkermansia muciniphila increases whole-body energy expenditure and fecal energy excretion in diet-induced obese mice. Gut Microbes 2020, 11, 1231-1245, doi:10.1080/19490976.2020.1737307.
  3. Plovier, H.; Everard, A.; Druart, C.; Depommier, C.; Van Hul, M.; Geurts, L.; Chilloux, J.; Ottman, N.; Duparc, T.; Lichtenstein, L., et al. A purified membrane protein from Akkermansia muciniphila or the pasteurized bacterium improves metabolism in obese and diabetic mice. Nat Med 2017, 23, 107-113, doi:10.1038/nm.4236.

Round 2

Reviewer 1 Report

The Authors appropriately revised the manuscript, by responding in detail to my concerns. 

In the current version, the paper is suitable for the publication in Foods.